# Vitamin D Supplementation and Its Impact on Mortality and Cardiovascular Outcomes: Systematic Review and Meta-Analysis of 80 Randomized Clinical Trials

**DOI:** 10.3390/nu15081810

**Published:** 2023-04-07

**Authors:** Antonio Ruiz-García, Vicente Pallarés-Carratalá, Miguel Turégano-Yedro, Ferran Torres, Víctor Sapena, Alejandro Martin-Gorgojo, Jose M. Martin-Moreno

**Affiliations:** 1Lipids and Cardiovascular Prevention Unit, Pinto University Health Center, 28320 Madrid, Spain; antoniodoctor@gmail.com; 2Department of Medicine, European University of Madrid, 28670 Madrid, Spain; 3Health Surveillance Unit, Castellón Mutual Insurance Union, 12003 Castellón, Spain; 4Department of Medicine, Universitat Jaume I, 12006 Castellón, Spain; 5Aldea Moret Health Center, 10195 Cáceres, Spain; tureyedro@hotmail.com; 6Biostatistics Unit, Medical School, Universitat Autònoma de Barcelona Bellaterra, 08193 Barcelona, Spain; ferran.torres@uab.cat (F.T.); victor.sapena@uab.cat (V.S.); 7STI/Dermatology Department, Madrid City Council, 28014 Madrid, Spain; alejandromartingorgojo@aedv.es; 8Department of Preventive Medicine and Public Health, University of Valencia, 46010 Valencia, Spain; 9Biomedical Research Institute INCLIVA, Clinic University Hospital, 46010 Valencia, Spain

**Keywords:** vitamin D, all-cause mortality, cardiovascular mortality, myocardial infarction, stroke, heart failure, major adverse cardiovascular events, systematic review, meta-analysis

## Abstract

Background: The impact of vitamin D supplementation on cardiovascular outcomes and mortality risk reduction remains unclear due to conflicting study findings. Methods: We conducted a systematic review and meta-analysis of randomized controlled trials (RCTs), published between 1983 and 2022, that reported the effect of vitamin D supplementation in adults versus placebo or no treatment on all-cause mortality (ACM), cardiovascular mortality (CVM), non-cardiovascular mortality (non-CVM), and cardiovascular morbidities. Only studies with a follow-up period longer than one year were included. The primary outcomes were ACM and CVM. Secondary outcomes were non-CVM, myocardial infarction, stroke, heart failure, and major or extended adverse cardiovascular events. Subgroup analyses were performed according to low-, fair- and good-quality RCTs. Results: Eighty RCTs were assessed, including 82,210 participants receiving vitamin D supplementation and 80,921 receiving placebo or no treatment. The participants’ mean (SD) age was 66.1 (11.2) years, and 68.6% were female. Vitamin D supplementation was associated with a lower risk of ACM (OR: 0.95 [95%CI 0.91–0.99] *p* = 0.013), was close to statistical significance for a lower risk of non-CVM (OR: 0.94 [95%CI 0.87–1.00] *p* = 0.055), and was not statistically associated with a lower risk of any cardiovascular morbi-mortality outcome. Meta-analysis of low-quality RCTs showed no association with cardiovascular or non-cardiovascular morbi-mortality outcomes. Conclusions: The emerging results of our meta-analysis present evidence that vitamin D supplementation appears to decrease the risk of ACM (especially convincing in the fair- and good-quality RCTs), while not showing a decrease in the specific cardiovascular morbidity and mortality risk. Thus, we conclude that further research is warranted in this area, with well-planned and executed studies as the basis for more robust recommendations.

## 1. Introduction

Vitamin D plays a critical role in maintaining and improving health in the musculoskeletal system by regulating calcium and phosphorus metabolism. In addition to its function in the musculoskeletal system, vitamin D may be relevant to the immune, dermatological, metabolic, or cardiovascular systems [1].

The vitamin D receptor and the enzyme 25-hydroxyvitamin D-1α-hydroxylase are expressed in many body tissues [2]. As an active metabolite of vitamin D, 1,25-dihydroxy-vitamin D (1,25(OH)_2_D) has anti-inflammatory, antiproliferative, antioxidant, and immunomodulatory effects. While there is biological plausibility between vitamin D supplementation and decreased mortality, the evidence of the impact of vitamin D supplementation and the reduction of cardiovascular disease (CVD) mortality is inconsistent [2].

Some observational studies suggest that vitamin D supplementation may improve the lipid profile and that there is an inverse association with cardiovascular events and mortality risk [3,4,5]. However, the results of the prospective cohort studies and randomized controlled trials (RCTs) are controversial, and doubts remain regarding their effects on coronary disease, stroke, all-cause mortality (ACM), and cardiovascular mortality (CVM) [4,5,6,7,8,9,10]. Some studies suggest that elevated levels of vitamin D lower the risk of coronary heart disease [4], others indicate a non-linear decrease in mortality risk as circulating vitamin D increases [5], and most reviews do not show that vitamin D supplementation decreases CVM or CVD [6,7,8,9,10].

Due to the conflicting evidence, some limitations of previous reviews, and the emergence of new trials evaluating the effect of vitamin D supplementation on mortality, the present systematic review and meta-analysis aimed to assess the impact of vitamin D supplementation on ACM, CVM, non-CVM, and CVD in adults.

## 2. Materials and Methods

### 2.1. Protocol and Guidance

The systematic review and meta-analysis followed Preferred Reporting Items for Systematic Reviews and Meta-Analyses (PRISMA) guidelines [11]. The protocol was registered in the International Prospective Register of Systematic Reviews (PROSPERO), registration number CRD42023393472.

### 2.2. Intervention, Control and Outcome Measures

The intervention was operationally defined as vitamin D supplementation with or without calcium. Vitamin D could be administered orally as vitamin D3 (cholecalciferol, [25(OH)D3]), vitamin D2 (ergocalciferol [25(OH)D2]), calcifediol or calcidiol (25-hydroxycholecalciferol or 25-hydroxyvitamin D [25(OH)D]), alfacalcidol (1α-hydroxy-vitamin D [1α(OH)D]), eldecalcitol, or calcitriol (1,25-dihydroxy-vitamin D [1,25(OH)_2_D]).

The control was no intervention or placebo with or without calcium, or another placebo drug in the control group if used equally in the intervention group. Outcome data at the end of the follow-up period of the RCTs were assessed.

Primary outcomes were ACM (deaths from any causes) and CVM (deaths due to CVD). Secondary outcomes were non-CVM (deaths from any causes except CVD), nonfatal myocardial infarction (MI), nonfatal stroke (hemorrhagic or ischemic), heart failure (HF), the composite endpoint of major adverse cardiovascular events (MACE) including CVM, nonfatal and nonfatal MI or stroke, and the extended MACE composite endpoint including MACE or coronary revascularization procedures (percutaneous coronary intervention or coronary artery bypass grafting).

### 2.3. Data Sources and Search Strategy

RCTs published in English from 1983 to 31 December 2022 (including online) that showed information on ACM or CVD morbidity and mortality were reviewed. The references of the included studies were screened.

RCTs were searched for in the EMBASE, MEDLINE, the Cochrane Library, and Google Scholar using the following search terms: “Vitamin D”, “Calcitriol”, “Calcifediol”, “Cholecalciferol”, “25-Hydroxyvitamin D 2”, “Ergocalciferols”, “calcium and vitamin D”, “cardiovascular disease”, “cardiovascular risk factors”, “coronary heart disease”, “myocardial infarction”, “stroke”, “heart failure”, and “morbidity and/or mortality” (Appendix A). Additional RCTs were identified by searching the reference lists of the included RCTs and systematic reviews, meta-analyses, and health technology assessment reports. Literature reviews or letters, comments, or animal studies were excluded. Figure 1 shows the search strategy in accordance with PRISMA guidelines [11].

### 2.4. RCTs Selection

RCTs that included at least 50 adult participants (≥18 years) during a follow-up period of at least one year and that also met the following eligibility criteria were selected:

Inclusion criteria: The intervention group received periodic oral vitamin D supplementation with or without calcium. The control arm group received a placebo with or without calcium or no treatment. The calcium or other placebo drugs had to be the same dosage in both groups.

Exclusion criteria: RCTs involving a single dose of vitamin D were excluded. RCTs were also excluded if the study subjects suffered from certain clinical conditions or comorbidities, namely if they were pregnant or lactating women, frail elderly, patients with acute coronary syndrome, stroke, COVID-19, HIV, or an estimated glomerular filtration rate < 30 mL/min/1.73 m^2^, patients in critical condition, and inpatients.

Two authors reviewed the title and abstract of full-text articles to assess study eligibility, determining whether the RCT met the inclusion criteria, whether the exclusion criteria were present, and whether any of the endpoints were reported. Disagreements related to study eligibility were resolved by consensus between the two, or by consulting a third author.

### 2.5. Data Extraction

Information from the selected RCTs was reviewed by two reviewers and checked against information available on ClinicalTrials.gov (https://clinicaltrials.gov/ct2/home; accessed on 13 January 2023) to independently assess and rate the methodological quality and strength of evidence of the RCTs. Two independent investigators used a standard data extraction form to extract data from the included RCTs. If RCTs had more than two comparison arms, data from intervention arms that were previously separated were pooled.

### 2.6. RCT Quality Assessment

To assess the risk of bias in the RCTs, the following five bias domains established by the “RoB 2” application [12] of the Cochrane Collaboration [13] were considered: selection (derived from the randomization process or concealment of allocation), performance (blinding in the allocation of participants or deviations from planned interventions), detection (blinding of outcome assessment), attrition (in outcome measurement or incomplete outcomes), and reporting (industry bias or selective reporting of results).

The overall quality assessment of the RCTs was defined as good, fair, or low according to whether they had a low, unclear, or high overall risk of bias. The RCTs were considered good quality if they had a low risk of bias in all domains or a single equivocal risk. They were classified as fair-quality RCTs if they had some limitations in several domains but had no high risk of bias. Finally, low-quality RCTs had an increased risk of bias in at least one domain or had some limitations in multiple domains. Two authors independently performed the critical quality appraisal of the RCTs. Disagreements in study quality ratings were resolved by mutual consensus or by consulting a third author in the case of a dispute.

Two authors extracted data independently and in a standardized manner. One author recorded in a structured form the main characteristics of the selected RCTs, including the number of participants in the intervention and control arms at the randomization of the RCT, the mean age and baseline characteristics of the study subjects, the mean treatment time and follow-up, the formulation and daily dose of the vitamin D used and of the placebo, and the data from the results. A second author checked that there were no errors in the inclusion of the data and that all were complete and accurate.

In the case of duplicate publications and supplementary papers from a primary RCT, information yield was maximized by evaluating all the available data simultaneously. If doubts arose, priority was given to the publication that reported more data, had a longer follow-up, or was more recent, and, in the case of equality, the one of better quality was prioritized.

### 2.7. Statistical Analyses

Meta-analyses were performed when at least three RCTs were available. The statistical analyses were carried out using the exact method proposed by Liu et al. [14] to use information from all the studies in this mixed zero-events meta-analysis, i.e., single-arm and double-arm zero-events studies with no zeros in the total count [15]. In this setting, this method allows the estimation of pooled odds ratios (ORs) and 95% confidence intervals (95%CI) without the need for artefact continuity corrections [16]. We used the R package “gmeta” to perform the exact method for this meta-analysis [17]. The heterogeneity of the results across the RCTs was assessed using the *I*^2^ statistic, which evaluates the percentage of between-study variance due to study heterogeneity versus sampling error, ranging from 0.0% (no heterogeneity) to 100% (high heterogeneity); values of 25%, 50%, and 75% have been suggested as benchmarks of low, moderate, and high heterogeneity [18]. Potential publication bias was considered significant when *p*-value (*p*) < 0.05 in either Begg’s test [19] or Egger’s test [20] when more than ten articles were included [21].

### 2.8. Subgroup Analyses

A subgroup analysis was performed to assess interactions according to the quality of the RCTs assessed according to RoB 2 [12], comparing the primary outcomes for ACM and CVM between low-, fair-, and good-quality RCTs (at risk of high, unclear, and low bias, respectively).

## 3. Results

The initial literature search yielded 4358 articles. After selecting titles and abstracts according to the inclusion and exclusion criteria, 361 RCTs were deemed suitable for evaluation, and 3997 were excluded. Following their full-text assessment, 281 RCTs were not included in the meta-analysis for the following exclusion reasons: 14 RCTs with fewer than 50 participants; 82 RCTs with less than one year of follow-up; 6 RCTs that included a single dose of vitamin D; 20 RCTs that assessed other drugs or added an intervention in either arm; 28 RCTs that included vitamin D in the control arm; 28 RCTs that included study subjects suffering from exclusionary clinical conditions or comorbidities; 12 RCTs without data/outcomes related to this review; 5 RCTs with other causes for exclusion (2 RCTs not written in English; 1 RCT with child participants; 2 RCTs with other formulations of vitamin D or non-oral route of administration), 4 retracted RCTs; and 82 RCTs whose results were duplicated in other RCTs included in the present study. The PRISMA flowchart of the search and selection of RCTs is shown in Figure 1. Therefore, 80 RCTs [22,23,24,25,26,27,28,29,30,31,32,33,34,35,36,37,38,39,40,41,42,43,44,45,46,47,48,49,50,51,52,53,54,55,56,57,58,59,60,61,62,63,64,65,66,67,68,69,70,71,72,73,74,75,76,77,78,79,80,81,82,83,84,85,86,87,88,89,90,91,92,93,94,95,96,97,98,99,100,101] were included in the present meta-analysis, of which 35 [22,23,24,25,26,27,28,29,30,31,32,33,34,35,36,37,38,39,40,41,42,43,44,45,46,47,48,49,50,51,52,53,54,55,56] were at low risk of bias, 34 [57,58,59,60,61,62,63,64,65,66,67,68,69,70,71,72,73,74,75,76,77,78,79,80,81,82,83,84,85,86,87,88,89,90] were at fair risk of bias, and 11 [91,92,93,94,95,96,97,98,99,100,101] were at high risk of bias.

The 80 RCTs [22,23,24,25,26,27,28,29,30,31,32,33,34,35,36,37,38,39,40,41,42,43,44,45,46,47,48,49,50,51,52,53,54,55,56,57,58,59,60,61,62,63,64,65,66,67,68,69,70,71,72,73,74,75,76,77,78,79,80,81,82,83,84,85,86,87,88,89,90,91,92,93,94,95,96,97,98,99,100,101] included 163,131 participants (82,210 in the intervention arm; 80,921 in the control arm), of whom 68.6% were women. The mean (SD) and median (IQR) age of participants were 66.1 (11.2) years and 66.7 (60.3–74.9) years, respectively. The age range was from 36.7 to 85.2 years. The mean (SD) and median (IQR) of the intervention duration were 2.2 (1.4) years and 2.0 (1.0–3.0) years, respectively. The mean (SD) and median (IQR) of the follow-up period were 2.3 (1.5) years and 2.0 (1.0–3.0) years, respectively.

The characteristics of the good-, fair-, and low-quality RCTs are outlined in Appendix A. The baseline characteristics of the participants from the good-, fair-, and low-quality RCTs are outlined in Appendix A summarizes the RCTs included according to the analysis of the outcomes.

The 35 good-quality RCTs [22,23,24,25,26,27,28,29,30,31,32,33,34,35,36,37,38,39,40,41,42,43,44,45,46,47,48,49,50,51,52,53,54,55,56] included 123,804 participants (62,631 in the intervention arm; 61,173 in the control arm), of whom 68.6% were women. The mean (SD) and median (IQR) of both the intervention duration and the follow-up period were 2.5 (1.7) years and 2.0 (1.0–3.0) years, respectively. Cholecalciferol (from 400 IU to 12,695 IU daily) was used in 30 studies, ergocalciferol (from 200 IU to 1429 IU daily) was used in 4 studies, 1.0 mcg daily of alfacalcidol was used in 1 study, and eldecalcitol (0.75 mcg daily) was used in another. Calcium (from 400 to 8000 mg) was added in the intervention arm in 11 studies and the control arm in 6 studies. Eicosapentaenoic acid (460 mg daily) and docosahexaenoic acid (380 mg daily) were added in 50% of participants in both comparison arms of two studies.

The 34 fair-quality RCTs [57,58,59,60,61,62,63,64,65,66,67,68,69,70,71,72,73,74,75,76,77,78,79,80,81,82,83,84,85,86,87,88,89,90] included 38,038 participants (18,901 in the intervention arm; 19,137 in the control arm), of whom 68.6% were women. The mean (SD) and median (IQR) of both the intervention duration and the follow-up period were 2.3 (1.3) years and 2.0 (1.0–3.0) years, respectively. Cholecalciferol (from 400 IU to 4000 IU daily) was used in 28 studies, ergocalciferol (from 822 IU to 7143 IU daily) in 3, alfacalcidol (0.75 mcg daily) in 1, and calcitriol (0.43 mcg and 0.50 mcg daily) in 2 studies. Calcium (from 150 mg to 1500 mg daily) was added in the intervention arm of 19 studies and the control arm of 9 studies.

The 11 low-quality RCTs [91,92,93,94,95,96,97,98,99,100,101] included 1289 participants (678 in the intervention arm; 611 in the control arm), of whom 74.6% were women. The mean (SD) and median (IQR) of the intervention duration were 1.2 (0.5) years and 1.0 (1.0–1.5) years, respectively. The mean (SD) and median (IQR) of the follow-up period were 1.4 (0.8) years and 1.0 (1.0–1.5) years, respectively. Cholecalciferol (from 300 IU to 12,186 IU daily) was used in 10 studies, and ergocalciferol (822 IU daily) in 1 of them. Calcium (from 500 mg to 1200 mg daily) was added in the intervention arm of 8 studies and the control arm of 4 studies.

### 3.1. Primary Outcomes

#### 3.1.1. All-Cause Mortality (ACM)

Meta-analysis of all the RCTs [22,23,24,25,26,27,28,29,30,31,32,33,34,35,36,37,38,39,40,41,42,43,44,45,46,47,48,49,50,51,52,53,54,55,56,57,58,59,60,61,62,63,64,65,66,67,68,69,70,71,72,73,74,75,76,77,78,79,80,81,82,83,84,85,86,87,88,89,90,91,92,93,94,95,96,97,98,99,100,101] showed that vitamin D supplementation reduced the ACM risk (OR 0.95 [95%CI 0.93 to 0.99], (*p* = 0.013), *I*^2^ = 0%). Subgroup analysis of the fair- and good-quality RCTs [22,23,24,25,26,27,28,29,30,31,32,33,34,35,36,37,38,39,40,41,42,43,44,45,46,47,48,49,50,51,52,53,54,55,56] showed that the pooled effect of vitamin D supplementation also decreased the ACM risk (OR 0.95 [95%CI 0.91 to 0.99], (*p* = 0.011) *I*^2^ = 0%). However, this effect bordered on statistical significance when the fair-quality RCTS [57,58,59,60,61,62,63,64,65,66,67,68,69,70,71,72,73,74,75,76,77,78,79,80,81,82,83,84,85,86,87,88,89,90] (OR 0.93 [95%CI 0.87 to 1.00], (*p* = 0.066) *I*^2^ = 0%) or the good-quality RCTs [22,23,24,25,26,27,28,29,30,31,32,33,34,35,36,37,38,39,40,41,42,43,44,45,46,47,48,49,50,51,52,53,54,55,56] (OR 0.96 [95%CI 0.91 to 1.00], (*p* = 0.067) *I*^2^ = 0%) were analyzed separately. Vitamin D supplementation was not associated with a lower ACM risk if only the low-quality RCTs [91,92,93,94,95,96,97,98,99,100,101] were analyzed (OR 1.22 [95%CI 0.67 to 2.29], (*p* = 0.510) *I*^2^ = 0%) (Figure 2).

#### 3.1.2. Cardiovascular Mortality (CVM)

Thirty-eight RCTs [24,26,28,31,33,34,36,38,42,45,46,47,48,49,50,51,53,55,56,58,60,70,75,76,77,78,79,80,81,82,86,90,91,94,96,97,98,99,101] with 110,424 participants (55,997 in the intervention arm; 54,427 in the control arm) were included, of whom 68.0% were female. The mean (SD) duration of the intervention and follow-up period was 2.02 (1.68) years. The mean (SD) age was 62.6 (10.4), ranging from 36.7 to 79.3 years.

Meta-analysis of all the RCTs showed no association between vitamin D supplementation and lower CVM risk (OR 1.00 [95%CI 0.92 to 1.08], (*p* = 0.986), *I*^2^ = 0%). Subgroup analysis also showed no significant differences between the comparison groups in the low-quality RCTs [91,94,96,97,98,99,101] (OR 1.06 [95%CI 0.37 to 3.05] (*p* = 0.900) *I*^2^ = 0%), fair-quality RCTs [58,60,70,75,76,77,78,79,80,81,82,86,90] (OR 1.28 [95%CI 0.55 to 3.06], (*p* = 0.565) *I*^2^ = 0%), or good-quality RCTs [24,26,28,31,33,34,36,38,42,45,46,47,48,49,50,51,53,55,56] (OR 1.00 [95%CI, 0.92 to 1.08], (*p* = 0.956) *I*^2^ = 0%). The pooled analysis of the fair- and good-quality RCTs [24,26,28,31,33,34,36,38,42,45,46,47,48,49,50,51,53,55,56,58,60,70,75,76,77,78,79,80,81,82,86,90] showed that vitamin D supplementation was not associated with lower CVM risk (OR 1.00 [95%CI 0.92 to 1.08], (*p* = 0.994) *I*^2^ = 0%) (Figure 3A).

### 3.2. Secondary Outcomes

#### 3.2.1. Non-Cardiovascular Mortality (Non-CVM)

The same RCTs analyzed for CVM outcomes were included.

The pooled effect of vitamin D supplementation in reducing the risk of non-CVM was close to statistical significance in the analysis of all the RCTs (OR 0.94 [95%CI 0.88 to 1.00], (*p* = 0.055), *I*^2^ = 0%) when analyzing the fair- and good-quality RCTs [24,26,28,31,33,34,36,38,42,45,46,47,48,49,50,51,53,55,56,58,60,70,75,76,77,78,79,80,81,82,86,90] (OR 0.94 [95%CI 0.87 to 1.00], (*p* = 0.053), *I*^2^ = 0%), and when analyzing only the good-quality RCTs [24,26,28,31,33,34,36,38,42,45,46,47,48,49,50,51,53,55,56] (OR 0.93 [95%CI 0.87 to 1.00], (*p* = 0.051), *I*^2^ = 0%). Subgroup analysis showed no significant differences between the comparison groups in the low-quality RCTs [91,94,96,97,98,99,101] (OR 1.06 [95%CI 0.37 to 3.05] (OR 1.15 [95%CI 0.26 to 5.55], (*p* = 0.847) *I*^2^ = 0%), and the fair-quality RCTs [58,60,70,75,76,77,78,79,80,81,82,86,90] (OR 1.01 [95%CI 0.57 to 1.78], (*p* = 0.974) *I*^2^ = 0%) (Figure 3B).

#### 3.2.2. Myocardial Infarction (MI)

Fourteen RCTs [24,28,29,36,37,45,48,56,65,70,78,83,84,91] with 83,053 participants (41,982 in the intervention arm; 41,071 in the control arm) were included, of whom 72.7% were female. The mean (SD) durations of the intervention and follow-up period were 3.61 (1.94) years and 3.78 (1.94) years, respectively. The mean (SD) age was 69.2 (6.7), ranging from 58.3 to 79.3 years.

Meta-analysis of all the RCTs showed no association between vitamin D supplementation and lower MI risk (OR 1.00 [95%CI 0.91 to 1.09], (*p* = 0.960), *I*^2^ = 0%). Subgroup analysis showed no significant differences between the comparison groups in the fair-quality RCTs 65,70,78,83,84] (OR 1.01 [95%CI 0.45 to 2.29], (*p* = 0.989) *I*^2^ = 0%) or the good-quality RCTs [24,28,29,36,37,45,48,56] (OR 1.00 [95%CI, 0.91 to 1.09], (*p* = 0.959) *I*^2^ = 0%) (Figure 4A).

#### 3.2.3. Stroke

Thirteen RCTs [24,28,29,36,37,45,48,56,65,70,78,84,91] with 82,640 participants (41,773 in the intervention arm; 40,867 in the control arm) were included, of whom 72.8% were female. The mean (SD) durations of the intervention and follow-up period were 3.74 (1.96) years and 3.92 (1.94) years, respectively. The mean (SD) age was 69.7 (6.7), ranging from 58.3 to 79.3 years.

Meta-analysis of all the RCTs showed no association between vitamin D supplementation and lower stroke risk (OR 1.04 [95%CI 0.95 to 1.13], (*p* = 0.430), *I*^2^ = 0%). Subgroup analysis showed no significant differences between the comparison groups in the fair-quality RCTs [65,70,78,84] (OR 1.41 [95%CI 0.57 to 3.73], (*p* = 0.460) *I*^2^ = 0%) or the good-quality RCTs [24,28,29,36,37,45,48,56] (OR 1.02 [95%CI, 0.94 to 1.12], (*p* = 0.600) *I*^2^ = 0%) (Figure 4B).

#### 3.2.4. Heart Failure (HF)

Four RCTs [28,36,45,78] with 47,903 participants (23,990 in the intervention arm; 23,913 in the control arm) were included, of whom 90.6% were female. The mean (SD) duration of the intervention and follow-up period was 4.70 (1.90) years. The mean (SD) age was 68.1 (6.5) years, ranging from 62.4 to 77.5 years.

The RCTs meta-analysis showed no association between vitamin D supplementation and lower heart failure risk (OR 0.94 [95%CI 0.84 to 1.06], (*p* = 0.305), *I*^2^ = 54.0%). Subgroup analysis showed no significant differences between the comparison groups in the good-quality RCTs [28,36,45] (OR 0.94 [95%CI, 0.83 to 1.05], (*p* = 0.276) *I*^2^ = 62.6%) (Figure 4C).

#### 3.2.5. MACE

Fourteen RCTs [24,28,29,31,36,37,45,48,56,65,70,78,84,91] with 29,587 participants (15,199 in the intervention arm; 14,388 in the control arm) were included, of whom 49.2% were female. The mean (SD) durations of the intervention and follow-up period were 3.69 (1.90) years and 3.85 (1.89) years, respectively. The mean (SD) age was 70.1 (6.64), ranging from 58.3 to 79.3 years.

Meta-analysis of all the RCTs showed no association between vitamin D supplementation and lower MACE risk (OR 1.00 [95%CI 0.95 to 1.06], (*p* = 1.00), *I*^2^ = 42.5%). Subgroup analysis showed no significant differences between the comparison groups in the fair-quality RCTs [65,70,78,84] (OR 1.23 [95%CI 0.66 to 2.30], (*p* = 0.516) *I*^2^ = 0%) or the good-quality RCTs [24,28,29,31,36,37,45,48,56] (OR 0.99 [95%CI, 0.94 to 1.05], (*p* = 0.830) *I*^2^ = 0%) (Figure 4D).

#### 3.2.6. Extended MACE

Three RCTs [48,56,78] with 54,403 participants (27,390 in the intervention arm; 27,013 in the control arm) were included, of whom 87.8% were female. The mean (SD) duration of the intervention and follow-up period was 5.27 (2.51) years. The mean (SD) age was 67.3 (0.8) years, ranging from 66.6 to 68.2 years.

Meta-analysis of all the RCTs showed no association between vitamin D supplementation and lower heart failure risk (OR 0.95 [95%CI 0.85 to 1.07], (*p* = 0.403), *I*^2^ = 0%). Subgroup analysis showed no significant differences between the comparison groups in the good-quality RCTs [48,56] (OR 0.96 [95%CI, 0.85 to 1.07], (*p* = 0.455) *I*^2^ = 0%) (Figure 4E).

## 4. Discussion

The possible association between low vitamin D levels and CVD, diabetes, cancer, and inflammatory disorders may be due to a large number of genes acting under the control of 1α,25(OH)_2_D. It may also be caused by the fact that the vitamin D receptor (VDR) and CYP27B1 1α-hydroxylation enzyme expression are located in most tissues and cells, including vascular endothelium and cardiomyocytes [102,103]. Measurements of vitamin D concerning the GC, CYP2R1, and DHCR7 genotypes in Mendelian randomization studies have shown associations of genotypes with vitamin D concentrations and all-cause mortality but not with CVM [104].

In addition to the classic risk factors that influence the development of CVD, some studies [3,4,5,6,7,8,9,10] have shown conflicting results regarding the effect of vitamin D supplementation on ACM, CVM, and CVD events in adults. Recent systematic reviews [105,106,107,108,109,110] of observational and cohort studies have reported an inverse association between vitamin D concentration and ACM, CVM, incident CVD events, and recurrent CVD events. In contrast, the meta-analysis conducted by Jaiswal et al. [111] reported an association between low blood levels of vitamin D and a higher incidence of MACE. Conversely, there was no significant association with ACM, MI, or HF risk. The evidence for the association with CVM and CVD events is weak without being able to prove its causality due to possible confounding factors, so intervention studies must support it.

Despite biological evidence and support from observational studies, recent meta-analyses and systematic reviews [112,113,114,115,116] of RCTs that have assessed the effect of vitamin D supplementation reported that it was not associated with a reduction in the risk of ACM, CVM, MI, stroke, HF, or MACE. A meta-analysis from the US Preventive Services Task Force (USPSTF) [117] included 77 fair- or good-quality RCTs and reported that vitamin D treatment did not affect mortality or CVD incidence. Another two recent meta-analyses from the USPSTF [118,119] reported that vitamin D use was not significantly associated with ACM, CVM, or CVD events. However, the meta-analysis by Bjelakovic et al. [10] stated that vitamin D supplementation decreased the risk of ACM in older adults.

The International Conference “Vitamin D Controversies” [1] held in Italy in 2017 summarized the extensive scientific progress on the therapeutic use and multiple biological actions of vitamin D in different tissues. During the conference, it was stated that much work remained to be done to create an even more secure knowledge base, especially the need to include more definitive RCTs to determine in which disorders vitamin D may be helpful therapeutically. In line with this suggested course of action, the present meta-analysis evaluates the RCTs according to their risk of bias, finding essential differences when ruling out low-quality RCTs.

In this updated meta-analysis, vitamin D supplementation showed results of increased ACM if the analysis included only low-quality RCTs. Conversely, when the meta-analysis was conducted including the 69 fair- and good-quality RCTs, it was associated with a reduction in ACM. On the other hand, vitamin D supplementation did not show a decreased risk of non-CVM if the meta-analysis included all the RCTs. However, it was still associated with a decrease in the risk of non-CVM if the meta-analysis was carried out with both the good-quality RCTs and the fair- and good-quality RCTs.

The present findings have the potential to provide insights into the divergent nature of the existing scientific literature assessing the impact of vitamin D supplementation on health outcomes. Furthermore, it warrants consideration of whether excluding substandard randomized controlled trials (RCTs) is necessary to ensure that recommendations are solely derived from robust investigations.

Although we strove to gather and scrutinize all the available evidence as rigorously as possible to meet the study objectives, we acknowledge that this review has several limitations. First, the effect of vitamin D supplementation on health outcomes was not analyzed according to the characteristics of the study populations (e.g., age, gender, or baseline vitamin D levels). Since we cannot know the vitamin D history of the participants in each RCT, and many of the conditions investigated may have long-term causes (risk factors accumulate over time), perhaps some protection will have accrued through adequate vitamin D levels throughout their lives, instead of being based on a beneficial reinforcement in the periods analyzed. Second, the effect of vitamin D supplementation on potential non-CVM causes was not examined. Third, the risks or benefits of different doses or formulations of vitamin D were not addressed here. Fourth, the meta-analysis included RCTs with non-elderly study populations with a low risk of mortality or CVD events. These factors limit the ability to detect the effect of vitamin D supplementation on the incidence and risk of mortality or CVD events. Fifth, no seasonal sub-analysis was performed because RCTs with a minimum follow-up period of one year were included; therefore, all seasons were included. Finally, the large number of analyses makes it difficult to rule out false positives due to chance.

## 5. Conclusions

The currently available evidence of the effect of vitamin D supplementation is based on many RCTs. The evidence that vitamin D supplementation in adults is inversely associated with ACM and non-CVM is consistent in fair- and good-quality RCTs. However, considering all the available studies, vitamin D supplementation is not significantly associated with a reduction in CVM, MI, stroke, HF, MACE, or extended MACE. In view of the above, and based on the promising indications of potential cardiovascular risk reduction benefits associated with adequate vitamin D supplementation from more rigorous studies, we believe it is worthwhile to explore the subject further through well-planned and executed studies in order to inform optimal recommendations.

## Figures and Tables

**Figure 1 nutrients-15-01810-f001:**
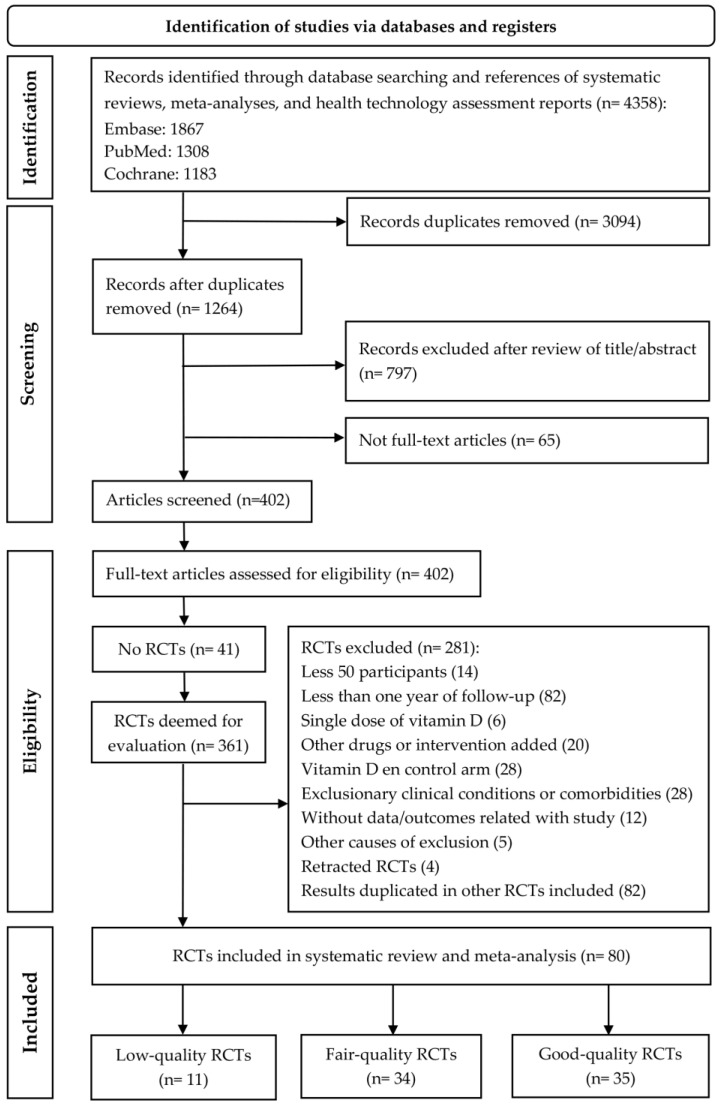
PRISMA flowchart of search and selection of RCTs.

**Figure 2 nutrients-15-01810-f002:**
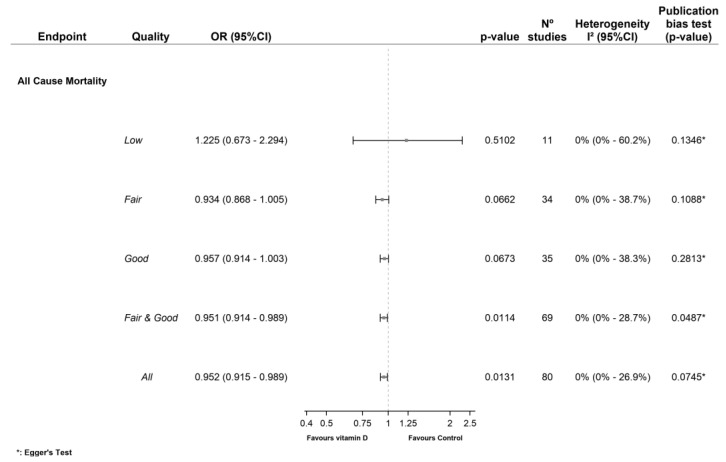
Forest plot of the effect of vitamin D supplementation on all-cause mortality.

**Figure 3 nutrients-15-01810-f003:**
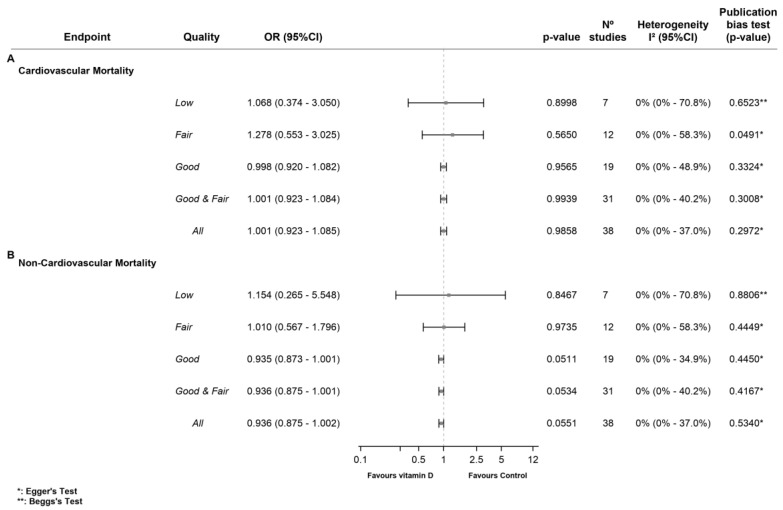
Effect of vitamin D supplementation on cardiovascular and non-cardiovascular mortality. (**A**) Forest plot of cardiovascular mortality. (**B**) Forest plot of non-cardiovascular mortality.

**Figure 4 nutrients-15-01810-f004:**
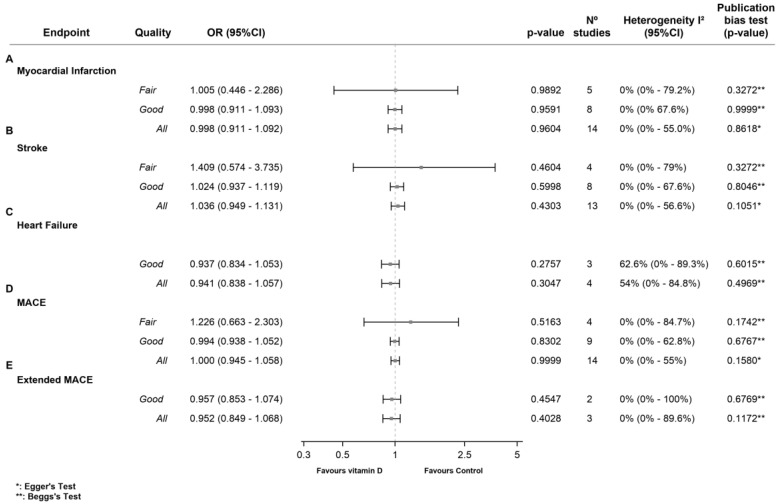
Forest plots of the effect of vitamin D supplementation on cardiovascular outcomes. (**A**) Effect of vitamin D supplementation on nonfatal myocardial infarction. (**B**) Effect of vitamin D supplementation on nonfatal stroke (haemorrhagic or ischemic). (**C**) Effect of vitamin D supplementation on heart failure. (**D**) Effect of vitamin D supplementation on the composite endpoint of major adverse cardiovascular events (MACE), including cardiovascular mortality, along with fatal and nonfatal myocardial infarction. (**E**) Effect of vitamin D supplementation on extended MACE, composite endpoint including MACE or coronary revascularization procedures (percutaneous coronary intervention or coronary artery bypass grafting).

## Data Availability

The data and the full statistical analyses are available upon request.

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
