# Peer review of "Vitamin D Supplementation and Its Impact on Mortality and Cardiovascular Outcomes: Systematic Review and Meta-Analysis of 80 Randomized Clinical Trials"

_nutrients, 2023, doi:10.3390/nu15081810_

Round 1
Reviewer 1 Report
The message of this paper is really important: that low quality RCTs can cause confusion and uncertainty in the outcome of a treatment. In sufficient number they can also skew the analysis in systematic reviews.
There are limitations to the study, but these are generally covered in the section beginning at line 390. However, it is suggested that some of these points might be expanded slightly e.g. baseline vitamin D level is not known / used and therefore some trials may have been using already replete participants for whom supplementation might not be expected to be effective.
Also, add that nothing is known of participant vitamin D history, and many of the conditions investigated can have long-term causes, or put another way risk factors build up over time. Maybe some protection accrues through suitable vitamin D status across the lifecourse, rather than a short-term boost.
Please increase the size of the text in the Forest plots.
Figure 3B is not discussed.
Lines 190-217 and lines 220-247 are identical. Delete one copy.
In this repeated section ranges of vitamin D supplementation are given for all categories of RCT quality. The units for cholecalciferol and ergocalciferol should surely be mcg (micrograms) and not mg (milligrams). Double check all other units. Same applies in Tables S3, S4, S5 – check units, and indicate that this is the daily dose.
Tables S3 and S4 have reversed orders of columns Vitamin D2 and 1α(OH)D. please make consistent.
Abstract – define non-CVM at first use (line 25).
Repetition of ‘with’, line 35.
Author Response
1.- The message of this paper is really important: that low quality RCTs can cause confusion and uncertainty in the outcome of a treatment. In sufficient number they can also skew the analysis in systematic reviews.
We appreciate the observation. Considering the issue carefully, we have decided to include only those RCTs with more than 1 year of follow-up, in order to reduce the problem of including conclusions derived from studies with insufficient follow-up.
2.- There are limitations to the study, but these are generally covered in the section beginning at line 390. However, it is suggested that some of these points might be expanded slightly e.g. baseline vitamin D level is not known / used and therefore some trials may have been using already replete participants for whom supplementation might not be expected to be effective.
Thanks for the remarks. After reviewing all the RCTs evaluated, the initial dose of intervention with vitamin D or Vitamin D/calcium is indicated (and explicit in the tables of the supplementary material). No reference is made in any of them to previous treatments, being an exceptional consideration that patients already treated previously and not described in the intervention sections of the RCTs are included, which is why we reflect this fact among the limitations of the study.
3.- Also, add that nothing is known of participant vitamin D history, and many of the conditions investigated can have long-term causes, or put another way risk factors build up over time. Maybe some protection accrues through suitable vitamin D status across the lifecourse, rather than a short-term boost.
Thanks. A paragraph in this sense is added in the new version of the manuscript.
4.- Please increase the size of the text in the Forest plots.
In this regard, it has not been possible to increase the size of the text due to the technical characteristics of the program used, as well as to ensure a correct layout in the integral visualization.
5.- Figure 3B is not discussed.
We appreciate the comment, but please note that a reference is made in the discussion of the manuscript.
6.- Lines 190-217 and lines 220-247 are identical. Delete one copy.
Thank you for noticing and letting us know. Redundant text has been deleted in the revised version of the manuscript
7.- In this repeated section ranges of vitamin D supplementation are given for all categories of RCT quality. The units for cholecalciferol and ergocalciferol should surely be mcg (micrograms) and not mg (milligrams). Double check all other units. Same applies in Tables S3, S4, S5 – check units, and indicate that this is the daily dose.
Thank you for the observation. This has now been consistently adapted in the revised version of the manuscript.
8.- Tables S3 and S4 have reversed orders of columns Vitamin D2 and 1α(OH)D. please make consistent.
Very sensible. Thanks. This has been modified.
9.- Abstract – define non-CVM at first use (line 25).
Acronyms have been systematically defined in their first entry in the text.
10.- Repetition of ‘with’, line 35.
Deleted in the new version of the manuscript
Reviewer 2 Report
The paper by Antonio Ruiz-García et al "Vitamin D Supplementation and Its Impact on Mortality and Cardiovascular Outcomes: Systematic Review and Meta-analysis of 156 Randomized Clinical Trials" has many aspects of concern.
The work presents a number of methodological difficulties. The abstract is not well written. Authors have analyzed many works making great effort and analysis of the literature but the results do not come to any conclusion. It is not possible to compare works were supplementation of vitamin D lasts only eight weeks; any cardiovascular effect o risk and vitamin D could be detected.
Doses of vitamin D are the most disparate and confusing and therefore are not comparable with each other.
The works should be collected at least for doses of vitamin D and for period of administration so, at presente, Authors can not drive any conclusions. It is not possible to compare any effect between vitamin D supplementations of eight weeks or of 10 years putting them all together.
Since Authors have done an important literature search, I suggest stratifying analysis by dosages of vitamin D and by period of intake
Author Response
The paper by Antonio Ruiz-García et al "Vitamin D Supplementation and Its Impact on Mortality and Cardiovascular Outcomes: Systematic Review and Meta-analysis of 156 Randomized Clinical Trials" has many aspects of concern.
The work presents a number of methodological difficulties. The abstract is not well written. Authors have analyzed many works making great effort and analysis of the literature but the results do not come to any conclusion. It is not possible to compare works were supplementation of vitamin D lasts only eight weeks; any cardiovascular effect o risk and vitamin D could be detected.
Doses of vitamin D are the most disparate and confusing and therefore are not comparable with each other.
The works should be collected at least for doses of vitamin D and for period of administration so, at presente, Authors can not drive any conclusions. It is not possible to compare any effect between vitamin D supplementations of eight weeks or of 10 years putting them all together.
Since Authors have done an important literature search, I suggest stratifying analysis by dosages of vitamin D and by period of intake
We thank the reviewer for the time spent reading and assessing our manuscript and for the comments on it, and we understand the doubts that may have been raised, particularly regarding the inclusion of studies with a follow-up period estimated as insufficient to properly assess effects, considering the necessary induction period.
In our revised version, we have decided to eliminate all those RCTs with a follow-up period of less than 1 year, so that the current manuscript only analyzes 80 RCTs, despite which the essence of the results and conclusions does not change substantially.
In a nutshell, we continue to believe that it is worth exploring the topic further through well-planned and executed studies to inform optimal recommendations (including consistency and better stratification of vitamin D doses).
On the other hand, the first version of our manuscript and the revised version attached in this new submission have been reviewed by an English scientific expert and we hope that there will be no language communication problems in that regard.
Reviewer 3 Report
The authors present an extensive meta analysis of 156 trials and more than 180,000 subjects dealing with vitamin D supplementation and all cause or CV mortality
They did not find a significant relationship for all cause mortality but limiting the study to good quality trials, a modest beneficial effect was found – low quality trials generated the non-significant opposite effect
No benefit was seen for CV mortality or subgroups of CV diseases
There are already many such studies but now it includes also the most recent data and thus the overview is of potential relevance
There is one MAJOR ERROR in the units of vitamin d as it is not mg but IU for lines 194-5; 203-4 and 211-2 and maybe other places !!!!!!!!!!!
The other major limitation is that the authors did not report a separate analysis according to mean baseline serum 25OHD concentrations or according to age.
The study should only include RCTs lasting for more than 12 and preferably more than 3 yrs as it is unlikely that mortality would be influenced by short therapeutic intervention
Author Response
The authors present a large meta-analysis of 156 trials and more than 180,000 subjects dealing with vitamin D supplementation and all-cause or cardiovascular mortality.
They did not find a significant relationship for all-cause mortality, but by limiting the study to good-quality trials, a modest benefit was found: low-quality trials produced the opposite non-significant effect.
No benefit was observed for CV mortality or CV disease subgroups
There are already many such studies, but now it also includes the most recent data and therefore the overview is of potential relevance.
There is a MAJOR ERROR in the vitamin d units since they are not mg but IU for lines 194-5; 203-4 and 211-2 and maybe in other places !!!!!!!!!!!
The other important limitation is that the authors did not report a separate analysis according to mean baseline serum 25OHD concentrations or according to age.
The study should only include RCTs lasting longer than 12 and preferably longer than 3 years, as mortality is unlikely to be influenced by a short therapeutic intervention.
We thank the reviewer for the time dedicated to the thorough examination and his/her comments on it. We understand the doubts that have been raised in the first version of our manuscript, particularly regarding the inclusion of studies with a follow-up period probably insufficient to properly assess effects, considering the necessary induction period.
In our revised version, we have decided to eliminate all those RCTs with a follow-up period of less than 1 year, so that the current manuscript only analyzes 80 RCTs (163,131 participants; 82,210 in the intervention arm), despite which the essence of the results and conclusions does not change substantially.
In this new version of the manuscript, the units (mg) have been corrected in the text and in the tables of the supplementary material to systematically adapt them to the SI units (IU).
Considering the above, and since all the RCTs included in the new version of the manuscript (n=80) the duration is greater than one year, and this minimized the risk of a dose-period bias, we still believe that it is worth exploring the topic further through well-planned and executed studies to inform optimal recommendations (including consistency and better stratification of vitamin D doses).
Round 2
Reviewer 2 Report
the authors did a good efford tho improve the paper so, it can be Accepted for publication in the present form